# Comparative Analysis of Temperature Fields in Railway Solid and Ventilated Brake Discs

**DOI:** 10.3390/ma14247804

**Published:** 2021-12-16

**Authors:** Aleksander Yevtushenko, Michal Kuciej, Piotr Grzes, Piotr Wasilewski

**Affiliations:** 1Department of Mechanics and Applied Computer Science, Faculty of Mechanical Engineering, Bialystok University of Technology (BUT), 45C Wiejska Street, 15-351 Bialystok, Poland; a.yevtushenko@pb.edu.pl (A.Y.); p.grzes@pb.edu.pl (P.G.); 2Frimatrail Frenoplast S.A., 15 Watykańska Street, 05-200 Majdan, Poland; pwasilewski@frimatrail-frenoplast.pl

**Keywords:** railway disc brake, frictional heating, moving heat source, temperature, finite element method

## Abstract

A new approach to numerical simulation using the finite element method (FEM) for the rotational motion of discs for railway vehicle disc brake systems was proposed. For this purpose, spatial models of transient heating due to the friction of such systems with solid and ventilated discs were developed. The performed calculations and the results obtained allowed justification of the possibility of simplifying the shape of the ventilated brake disc through elimination of ventilation channels. This contributes to a significant reduction in computational time, without compromising the accuracy of the results. The spatial and temporal temperature distributions in the ventilated and the solid disc of the same mass were analyzed. The share of energy dissipated due to convection and thermal radiation to the environment in relation to the total work done during a single braking was investigated. The maximum temperature values found as a result of computer simulations were consistent with the corresponding experimental results.

## 1. Introduction

Carrying out a computer simulation with the use of FEA of the transient temperature field of the brake consists of several steps. In addition to the precise determination of the operating parameters as well as the shape and dimensions of the friction elements, it requires simplifying assumptions while developing the calculation models. Modern computational units make it possible to include a number of interdependent quantities (e.g., coefficient of friction, thermophysical properties, heat transfer coefficient, hardness, average temperature of the nominal contact area, flash temperature, maximum temperature, etc.) in numerical calculations of the thermal problems of friction. These values correspond with the complex shapes of the brake pads and discs, and even entire assemblies of parts in the vicinity of the brake [1,2,3]. Such an approach, aimed at a comprehensive assessment of the temperature state, is based on a system of equations of thermal dynamics of friction and wear [4]. The basic problems of this system concern (1) the determination of the time profiles of pressure, velocity, and friction power density; (2) obtaining experimental and then analytical relations of the coefficients of friction and wear rate on temperature; (3) the experimental dependency of the temperature-dependent material properties; (4) the determination of the average temperature of the nominal contact area and the temperature of the real contact area (flash temperature) and eventually their sum—the maximum temperature [4].

Along with the increase in the complexity of the computational model of the process by generalizing the basic equations of the HDFW (heat dynamics of friction and wear) system to different braking modes and brake systems, the computation time may be significantly extended. In order to shorten that time, analytical or numerical models that do not take into account temperature changes in the circumferential direction (axisymmetric or 2D models) are used [5]. Omission of the circumferential coordinates allow us to obtain reliable results with a solid disc, associated with the pads in the shape of a ring segment. Then, by introducing the heat partition coefficient, the temperature is set separately for the disc (2D model) and the pads (3D model). In the case of a ventilated disc with an irregular shape of the nominal contact area (due to external contours and internal cuts), only 3D calculation models should be used to determine the temperature mode of the brake.

Changes in transient temperature fields of a passenger vehicle brake disc with exponentially increasing pressure and non-linearly decreasing disc angular velocity were analyzed in the article [6]. The cyclic heating of the surface of the friction ring (rubbing path) of the disc was modeled by means of a moving surface heat source with alternately changing areas of heating and convection cooling. The changes in the temperature, braking time, and heat flux densities on the working surface of the disc were presented. A similar approach was proposed in [5]. During the numerical simulation of a moving heat source, a program code developed by the authors was proposed, generating boundary conditions on the working surface of the disc in the MES MSC.Patran/MSC.Nastran environment. For this purpose, functions describing the change of the heat flux density heating the disc were developed separately for each of the finite elements uniformly distributed in the circumferential and radial direction. This approach provides an accurate representation of the frictional power density portion in the area of displacement of the pad relative to the stationary disc. A significant advantage of the proposed method is the possibility of adapting this technique to a ventilated disc, provided that its outer surface is consistent and homogenous in the area of heating. The disadvantage, however, is that it is not possible to perform calculations using the contact model of heating the brake, taking into account the simultaneous mutual influence of the pads and the disc on the temperature.

The approach proposed in [6] was adapted to the thermal-structural coupling model in order to determine not only the temperature field, but also to carry out transient stress analysis and to describe the phenomenon of hot spot formation [7]. The main purpose of the work was to develop a methodology for identifying thermal fatigue cracks. The calculations were carried out for the regular shapes of the solid disc and pads. It was shown that at a fixed point of the disc, the temperature evolution takes the form of cyclically repeating stages of rising and falling with each revolution of the wheel.

The temperature fields of the ventilated disc brake elements during a single braking were determined numerically in the software environment based on the ANSYS Workbench platform [8]. In the developed contact thermo-structural coupled computational model, the stopping time as well as the exponential increase in contact pressure and the nonlinear velocity change were known a priori before the analysis. This also includes the variable heat transfer coefficient from the surfaces of the pads and the disc. The simulations were carried out using the direct coupling method, which is an iterative calculation of temperature and stress fields. The temperature changes over time at selected points on the disc obtained by the calculations were consistent with the corresponding results obtained using thermocouples embedded in these points. The temperature time profiles on the working surface of the disc revealed distinct and regular oscillations, declining with the distance from the friction surface.

Transient temperature fields of the pad and ventilated disc were found using the finite element method, adapted in COMSOL Multiphysics^®^ [9]. Using the heat partition coefficient, simulations of frictional heating were carried out separately for the disc and the pads. Although the calculations of the temperature of the pads did not require interference with the standard tools available in the commercial FEM software environment, special modules (Mathematics and Deformed Geometry) were used to determine the temperature of the disc, to represent the movement of the heating area of a complex shape on the friction surface of the stationary disc. In fact, the change in the rotational velocity of the ventilated disc and the deformation of the surface layer of its geometric model were related, allowing for an accurate representation of the heat flux density proportional to the power of friction forces during braking.

The temperature fields of the disc and pads during a single braking of a rail vehicle were analyzed in the article [10]. The process of frictional heating during braking was simulated on a dynamometer test bench. When developing the calculation model, the ventilated disc was replaced with a solid disc with the same outer dimensions and mass. The same mass was maintained by increasing the thickness of the heated layer parallel to the friction surface as well as by simplifying the area of the disc-hub interface. In the contact area of the pad with the disc, perfect thermal friction conditions were assumed, i.e., the temperature of the opposite surfaces at each point was equal, and the sum of the heat flux densities directed from the friction surface to the inside of the disc and the pads was equal to the friction power density. In each of the ten tested variants of the single braking, a high consistency of the maximum disc temperature value, calculated by means of FEM and measured using thermocouples, was achieved. It was found that replacing a ventilated disc with a solid disc of the same mass during a single braking provides a sufficient level of accuracy in finding the temperature.

Currently, two types of brake discs are used in railway vehicles: solid and ventilated (in radial or tangential direction). The discs are manufactured as monoblocks or as individual friction rings. According to [11], taking into account the dimensions of the axle-mounted railway discs (typically 640 mm in diameter) and thicknesses from 80 (steel) to 110 mm (cast iron), high air pumping leads to aerodynamic loses and energy consumption. As a result, operating costs and the negative impact on the environment increase. In the above-mentioned work, the disc was developed with both radial vanes and circumferential pillars, the so-called radial vanes/pillared design. As was shown, such a structure of the disc does not differ from the traditional one in terms of the amount of energy dissipated by convection, and at the same time it shows much better (about 50%) equivalent aerodynamic (air pumping) losses. It was shown that a parameter that must be taken into account when designing discs is the ratio of convective power dissipation to aerodynamic power losses. Using the computational fluid dynamics (CFD) method, the changes during braking of the parameters, such as aerodynamic (pumping) power loss, convective-heat dissipation (power loss), and the aerodynamic efficiency ratio dependent on the angular velocity of the disc, were analyzed. It was emphasized that the commonly used discs for rail vehicles were produced for years without significant changes, before the introduction of the advanced CFD methods.

At a given time, there are many different approaches, methods, and simplifications in the numerical modeling of temperature fields of a rotating disc and stationary brake pads. Their analyses are provided in review articles [12,13]. Undoubtedly, one of the most important requirements when developing such models, apart from the accurate determination of the maximum temperature and flash temperature occurring in the real contact areas of the pads with the disc, the variability of friction coefficients and wear intensity, etc., was to shorten the computational time. Recent achievements in the modeling of disc temperature fields, the main goal of which was to develop a model that is universal and does not require significant time to obtain results, are presented in [14]. The authors proposed a calculation scheme that they described as a uniformly distributed heat source method, abbreviated as the UDHS (uniformly distributed heat source) method [14]. The method, unlike those in the literature, aimed at finding a way to obtain accurate results in a short time. The basic simplification assumption of the UDHS method consists of adopting a homogeneous distribution of heat flux intensity on the friction surface and approximating the mutual cyclic motion of the disc and the pad with the cosine function. The obtained results were verified on the basis of temperature measurements using thermocouples on a full-scale test bench.

A similar but more advanced approach, based on the Gaussian mixture function (GMHS—Gaussian mixture heat source Method), was proposed in [15]. The method, as in [14], is based on the Gaussian function approximation of the heat flux intensity distribution at a specific point on the working surface of the disc. At a constant angular velocity, these cycles were evenly distributed in time, and at a linearly decreasing velocity, the maximum value of the flux decreased with each subsequent cycle. It should be emphasized that this method has a significant simplification. At a given moment, the entire friction surface of the disc (in the circumferential and radial direction) was simultaneously heated with a heat flux of one and the same value.

A similar approach, albeit for the 2D axisymmetric model, simplifying the 3D ventilated disc model, was proposed in [16]. Calculations were performed for the braking parameters, disc dimensions, and material properties, as adapted from [17]. The function approximating the successive passes of the pad relative to the disc was developed so as to accurately map the temporal profile of the heat flux density (rectangular waveform), determined on the basis of the exponentially increasing pressure and the nonlinearly decreasing angular velocity of the disc. The calculated temperature values were compared with the corresponding measurement data from [17], and a very good agreement was obtained.

Initiated by an axisymmetric heat load, the spatial temperature field of the ventilated disc was determined numerically by means of FEM and CFD in [18]. The wheel-mounted brake disc R920K for the ER24PC locomotive was considered. First, distributions of the heat transfer coefficient were found in the FLUENT environment, and then the obtained results were used as boundary conditions for thermal analysis with the heat partition coefficient. It was found that the ventilation channels are heated after a certain time (a few seconds) from the beginning of the braking process. This delay in a temperature rise resulted in a small effect of convection cooling. Parallel laboratory tests were used to verify the calculations. The determined temperature evolution during braking agreed well with the data from the experimental studies.

The developed numerical model of the disc brake with axisymmetric heating of the ventilated disc [18] was then used to determine the thermal stress fields formed in the disc during emergency braking and during braking when riding downhill with a constant velocity [19].

In this paper, a comparative analysis of the temperature mode of the ventilated and solid discs of a rail vehicle was carried out, considering a single brake application. For this purpose, two techniques for modeling the rotational motion of the disc or the displacement of the pad relative to a fixed disc are proposed. The problems related to the optimization of the selection of a temporal step of such a size that would allow for an accurate mapping of the oscillating temperature changes at a specific point of the working surface of the disc are discussed in detail. During the calculations, the condition of equality of the total friction work found numerically and the initial kinetic braking energy obtained during experimental tests on a full-scale dynamometer test bench are verified. The calculated temperature values are shown to be consistent with the corresponding thermocouple measurement data.

## 2. Statement of the Problem

The subject of the study is the temperature field generated due to friction in a disc brake of a railway vehicle (Figure 1). The friction pair consists of the tangential vane type brake disc (640 mm × 110 mm), with the reduced ventilation and a set of organic composite brake pads typically used in passenger coaches.

During a single braking, the pressure p was the same at each point of the contact area, increasing linearly in time t from zero at the initial time t=0 to the value $p_{1}$ at $t=t_{1}$, then increasing stepwise to the nominal value $p_{s}$ and remaining at this value until the end of the process $t=t_{s}$:(1)$$p(t)=\{\begin{array}{c} p_{1}t/t_{1},\;0\leq t\leq t_{1},\\ p_{s}\;,\;t_{1}\leq t\leq t_{s}, \end{array}$$
where $p_{1}=0.5F_{1}/A_{a}$, $p_{s}=0.5F_{s}/A_{a}$, $F_{1}$—the value of the clamping force $t=t_{1}$, $F_{s}=1.05F_{1}$, and $A_{a}$—surface area of the nominal contact of the pad with the disc. The change of pressure in time (Equation (1)) corresponds to the following time profile of the angular velocity of the disc:(2)$$\omega (t)=\left\{ \begin{array}{l} \omega_{0}+(\omega_{1}-\omega_{0})t/t_{1},\;\;\;0\leq t\leq t_{1},\\ \omega_{1}+(\omega_{s}-\omega_{1})(t-t_{1})/(t_{s}-t_{1}),\;\;\;t_{1}<t<t_{s}. \end{array}\right.$$
where $\omega_{0}\equiv \omega (0)$, $\omega_{1}\equiv \omega (t_{1})$, and $\omega_{s}\equiv \omega (t_{s})$.

As a result of friction, heat is generated in the contact area between the pad and the disc, causing them to heat up. When developing the calculation model, the following assumptions were made to determine the brake temperature field:The pads are positioned symmetrically on both sides of the disc. Therefore, due to the existing load and geometric symmetry, only the half-thickness disc in combination with one pad is analyzed;The thermal contact of friction between the brake pads and the disc is perfect, i.e., the temperature in the contact area is the same, and the sum of the heat flux densities directed normally from the friction surface to the inside of each element is equal to the friction power density;The cooling of the free surfaces of the pads and the disc proceeds due to convection and thermal radiation to the surrounding air;Thermophysical properties of materials as well as friction and heat transfer coefficients do not change under the influence of temperature.

The parameters and quantities relating to the pad and the disc are denoted by the subscripts “p” and “d”, respectively, and the pad-disc friction pair is related to the cylindrical coordinate system of spatial variables (r,θ,z). On the basis of assumptions 1–3, the transient temperature field T(r,θ,z,t) of the brake was found from the solution of the following heat conduction equations of the parabolic type:(3)$$\frac{\partial^{2}T}{\partial r^{2}}+\frac{1}{r}\frac{\partial T}{\partial r}+\frac{1}{r^{2}}\frac{\partial^{2}T}{\partial \theta^{2}}+\frac{\partial^{2}T}{\partial z^{2}}=\frac{1}{k_{p}}\frac{\partial T}{\partial t},\;(r,\theta ,z)\in \Omega_{p},\;\;0<t\leq t_{s},$$
(4)$$\frac{\partial^{2}T}{\partial r^{2}}+\frac{1}{r}\frac{\partial T}{\partial r}+\frac{1}{r^{2}}\frac{\partial^{2}T}{\partial \theta^{2}}+\frac{\partial^{2}T}{\partial z^{2}}=\frac{1}{k_{d}}\left[\frac{\partial T}{\partial t}+\omega (t)\frac{\partial T}{\partial \theta }\right],\;(r,\theta ,z)\in \Omega_{d},\;\;0<t\leq t_{s},$$
where
(5)$$\Omega_{p}=\{(r,\theta ,z)\in R^{3}:\;r_{p}\leq r\leq R_{p},\;-\theta_{0}\leq \theta \leq \theta_{0},\;0\leq z\leq \delta_{p}\},$$
(6)$$\Omega_{d}=\{(r,\theta ,z)\in R^{3}:\;r_{d}\leq r\leq R_{d},\;-\pi \leq \theta \leq \pi ,\;0\leq z\leq \delta_{d}\},$$
$k_{p,d},r_{p,d}$ and $R_{p,d}$ are the coefficients of thermal diffusivity and the inner and outer radii of the elements, respectively, and $2\theta_{0}$—cover angle of the pad.

Taking into account assumption 4, in the area of contact of the pad with the disc,
(7)$$\Gamma =\{(r,\theta )\in R^{2}:\;r_{p}\leq r\leq R_{p},\;-\theta_{0}\leq \theta \leq \theta_{0},\;z=0\},$$

The following conditions should be met [20]:(8)$$T(r,\theta ,0^{+},t)=T(r,\theta ,0^{-},t),\;(r,\theta )\in \Gamma ,\;0<t\leq t_{s},$$
(9)$${K_{d}\frac{\partial T}{\partial z}|}_{z=0^{-}}-{K_{p}\frac{\partial T}{\partial z}|}_{z=0^{+}}=q(r,\theta ,t),\;(r,\theta )\in \Gamma ,\;0<t\leq t_{s},$$
where $K_{p,d}$—thermal conductivity of materials, and the friction power density was determined from the equation:(10)$$q(r,\theta ,t)=fp(t)r\omega (t),\;(r,\theta )\in \Gamma ,\;\;0<t\leq t_{s},$$
in which f—coefficient of friction, and p(t) and ω(t)—pressure (Equation (1)) and angular velocity (Equation (2)) time profiles, respectively.

Based on Assumption 2, the time profiles of the heat flux densities directed normally from the free surface $\Gamma_{p,d}$ of the pad and the disc to the environment were written as
(11)$$q_{p,d}^{diss}(t)=q_{p,d}^{conv}(t)+q_{p,d}^{rad}(t),\;\;0<t\leq t_{s},$$
where
(12)$$q_{p,d}^{conv}(t)=h(T_{a}-T),\;q_{p,d}^{rad}(t)=\varepsilon_{p,d}\sigma (T_{a}^{4}-T^{4}),$$
h—heat transfer coefficient, $\varepsilon_{p,d}$—emissivity of materials, $\sigma =5.67\cdot 10^{-8}\;W\;m^{-2}K^{-4}$—Stefan-Boltzmann constant, and $T_{a}$—ambient temperature.

Initially, the pad and the disc were at temperature $T_{0}$.

Taking into account the form of the friction power density q (Equation (10)), the time course of the friction work during braking W was determined from the following equation:(13)$$W(t)=2A_{a}\int_{0}^{t}\;q(r_{eq},0,\tau )d\tau ,\;\;0<t\leq t_{s},$$
where
(14)$$r_{eq}=\frac{2(R_{p}^{3}-r_{p}^{3})}{3(R_{p}^{2}-r_{p}^{2})},$$
$A_{a}$—surface area of contact of the pad on one side of the disc.

On the other hand, taking into account Equations (11) and (12), the changes in the braking time of the amount of heat $W_{p,d}^{diss}$, dissipated from the free surfaces to the environment, were determined from the equations:(15)$$W_{p,d}^{diss}(t)=2[W_{p,d}^{conv}(t)+W_{p,d}^{rad}(t)],\;\;\;\;\;\;0<t\leq t_{s},$$
where
(16)$$W_{p,d}^{conv}(t)=A_{p,d}\int_{0}^{t}\;q_{p,d}^{conv}(\tau )d\tau ,\;\;\;W_{p,d}^{rad}(t)=A_{p,d}\int_{0}^{t}\;q_{p,d}^{rad}(\tau )d\tau ,$$
$A_{p,d}$—half of the free surface areas $\Gamma_{p,d}$ of two pads and one disc.

## 3. Development of the 3D CAD Geometric Model and Generation of the Finite Element Mesh

The solution of the boundary value problem of heat conduction (Equations (3)–(10)) was obtained numerically using FEM, adapted in the COMSOL Multiphysics^®^ software package [21]. The calculations were carried out on the basis of the three models, with a different degree of simplification of the shape of the disc, consisting mainly of taking into account or not taking into account the ventilation channels of the disc. These are models where:The ribs forming the channels were replaced with a solid area of reduced thickness so as to keep the mass of the disc unchanged. In this model, the relative rotational movement of the disc and the pad takes place by assigning a variable velocity field at each point of the disc area at a fixed pad [10];The actual shape of the disc with ventilation channels was taken into account. The rotational movement of the disc was replaced by the movement of the pad relative to the stationary disc. This is done by changing the outer contours of the disc in the vicinity of the contact surface according to the simulated displacement of the pad;The solid disc from model I was considered, but with the relative displacement of the components according to model II.

One of the goals of this work was to carry out a comparative analysis of the brake temperature fields with a ventilated disc and a solid disc, representing a ventilated disc in a simplified form. A high convergence (relative maximum temperature difference did not exceeded 0.01%) of the results for the solid disc, obtained on the basis of the well-known and well-approved model I, was established with the corresponding temperature values obtained with the use of the new model III. This allowed us to be sure of the correct choice of the proposed modelling approach and the COMSOL Multiphysics^®^ tools applied in model III, and then II. Therefore, only the results obtained from model III (solid disc) and model II (ventilated disc) will be presented and discussed (Figure 2).

The 3D CAD geometric models of the tested braking system used to generate the finite element grids were created using the SOLIDWORKS^®^ software (Version 2020). The construction of the CAD geometric model of the disc required taking into account the specificity of assigning properties and parameters of the Deformed Geometry tools of the COMSOL Multiphysics^®^ program. In the geometric model of the pad, the wall inclinations were ignored due to the considerable distance from the friction surface (more than 10 mm). The backing plate, which strengthens the lining and allows it to be properly mounted in the holder, was replaced with the pad material. Holes were also omitted to improve the stability of the connection between the sheet and the pad. The brake pad holder was also not taken into account. Due to the considered short-term braking mode, all these simplifications did not affect the obtained results, but allowed us to reduce the total number of finite elements and thus significantly (several times) shortened the computational time. Considering the possible practical application of the proposed numerical FE models, this is of key importance.

The geometrical models of the ventilated and solid discs were transferred from SOLIDWORKS^®^ to COMSOL Multiphysics^®^ using and the Live Link™ tool (Version 5.3). At the same time, all modifications made in SOLIDWORKS^®^ could be constantly updated in COMSOL Multiphysics®. Then, a second-order finite element mesh (quadratic Lagrange) was generated. The overall final FE mesh was created in a few steps. An important criterion for changing the size of the element of the parts of the assembly was the expected high temperature gradients resulting from the frictional heating process corresponding with the properties of materials. It was necessary to include several times lower thermal conductivity of the pad compared to the steel disc. Therefore, after creating the 2D triangle elements on the plane friction surface (15 mm was the maximum element size), the sweep feature with a predefined distribution type (arithmetic sequence) was created. In model III, the number of the finite elements within the area of the solid disc was equal to 179 tetrahedral and 24,036 prism elements, and the pad was divided into 2048 prism elements. Whereas in model II, the disc consisted of 33,518 tetrahedral, 50 pyramid, 14,595 prism and 140 hexagonal elements. In this model, the pad consisted of 4669 tetrahedral and 2048 prism elements.

## 4. Modeling the Rotational Motion of the Disc

In the case of a solid disc with no shape changes in the circumferential direction, the numerical modeling of temperature fields in FEM-based programs usually requires the following:Separating the pad friction surface and thus defining the nominal contact area (heating area);Formulating the boundary conditions on each surface (heat flux density, convection cooling, thermal insulation, etc.);Defining a mathematical relationship for the velocity of each point of the disc during braking in order to obtain the effect of its rotation with respect to the stationary heating area.

The difficulty of describing the pad-disc frictional heating process arises when the disc is ventilated. In this case, its shape changes in the circumferential direction, caused by the free spaces alternating with elements constituting the ribs. This requires writing in-house program code to generate the function of changing the heat flux density separately for each element in the area of the surface of the rubbing path of the disc [5,22].

This article proposes the adaptation of special tools, available in the COMSOL Multiphysics^®^ software, to deform the geometry in such a way that it corresponds to the rotational motion of the ventilated disc in relation to the stationary pad. This task was carried out for the disc, specially divided into objects for this purpose. The contours of the isolated heating area of the disc with a complex shape, described by the boundaries of the pad’s friction surface, were deformed during braking. In spite of the rotation of the disc, the conditions of perfect thermal friction contact, with the fixed pad remaining in the initial position, were maintained. The described deformation occurred only for selected elements, constituting a solid layer of a certain thickness (the condition of the possibility of deformation is the continuity of the material). Due to the presence of ribs, the adjacent layer towards the inside of the disc, parallel to the contact surface, remained stationary (like the pad); see Figure 3a,b.

The problem formulated in this way required the use of specially adapted tools, available in the model for Deformed Geometry in COMSOL Multiphysics^®^ software. When describing the rotational motion of the disc contours, taking into account the heating area, the following transformation equations were used for the components $\left(b_{x},b_{y}\right)$ and $\left(b_{x^{\prime }},b_{y^{\prime }}\right)$ of any vector $\vec{b}$ for the rotation of the Cartesian coordinate system $Ox^{\prime }y^{\prime }$ by the angle θ with respect to the stationary orthogonal system Oxy [23,24]:(17)$$b_{x^{\prime }}=b_{x}\cos\theta +b_{y}\sin\theta ,\;b_{y^{\prime }}=-b_{x}\sin\theta +b_{y}\cos\theta ,$$
or
(18)$$b_{x}=b_{x^{\prime }}\cos\theta -b_{y^{\prime }}\sin\theta ,\;b_{y}=b_{x^{\prime }}\sin\theta +b_{y^{\prime }}\cos\theta .$$

Then, in the system Oxy, the rotation of the disc contours with a separate heating area, corresponding to the rotation of the lining relative to the stationary disc, was described by a vector $\vec{d}$ with the following components:(19)$$d_{x}=b_{x}-b_{x^{\prime }},\;d_{y}=b_{y}-b_{y^{\prime }},\;d_{z}=0.$$

During the calculations with the use of Equations (17)–(19), it should be remembered that the circumferential variable θ is not constant, but has a sense of the angular distance ϑ(t), which changes with time. According to Equation (2), the braking process consisted of two stages with linearly decreasing rotational velocity ω with constant, different at each stage, deceleration values. Therefore, in the COMSOL Multiphysics^®^ program, first the rotational velocity values were calculated from Equation (2) with time steps Δt=0.001 s, and then the appropriate values of the angular velocity ω(t) of the contact area Γ were determined on their basis:(20)$$\vartheta (t)=\int_{0}^{t}\;\omega (\tau )d\tau ,\;\;0<t\leq t_{s}.$$

The first calculations according to the above-described scheme showed that the temperature oscillations resulting from the displacement of the pad relative to the disc are so frequent that they require a minimum data reading step of no more than 0.002 s. Such a high concentration of data reading points for the braking process lasting several seconds leads to a significant increase in the computation time. In order to reduce this time, while maintaining the correct course of temperature evolution, it was decided to perform additional optimization calculations out of the COMSOL Multiphysics^®^ environment. For this purpose, a code for determining the angular velocity ω(t), $\;0<t\leq t_{s}$ was written in the Python programming and scripting language (Equation (2)). This made it possible to generate a table of values of $W_{p,d}^{diss}$ with a time step of 0.001 s. It should be noted that it can also be implemented on an ongoing basis in the program, using the tools for solving differential equations for Global ODEs (ordinary differential equations) and DAEs (differential-algebraic equations) (calculation of the angular distance) from the Mathematics module; however, after preliminary thermal analyses, it was found that the results obtained in this way were not accurate enough. Thus, in order to determine the angular distance ϑ (Equation (20)), the integration of the tabulated angular velocity ω was performed. The optimization of the temperature calculation time was performed using two time step values. A list of points was generated, such that the five time intervals before and after the left (leading) edge as well as the right (exit) edge of the pad had a time step of Δt=0.002 s, and the remaining ones had a step of Δt=0.01 s. A very small time step of Δt=0.00001 s was applied to create this list in the original, in-house Python code. In addition, the data format in COMSOL Multiphysics^®^ required writing the range in the form ($t_{0}$,Δt,$t_{end}$), where $t_{0}$ and $t_{end}$(the beginning and the end value of the time interval, respectively) had to be taken into account.

## 5. Numerical Analysis

Theoretical calculations were performed for the tangential vane type brake disc (640 mm × 110 mm) with the reduced ventilation and two organic composite brake pads (Figure 1). This friction pair was modeled using the solid disc (models I and III) and the ventilated disc (model II) with the same cover angle of the pad $2\theta_{0}=86.9{}^{\circ}$ at the ambient temperature $T_{a}=20\;{}^{\circ}C$ (Figure 2). The values of the thermophysical and mechanical constants of the materials as well as the dimensions of the pads and the disc shown in Table 1 were used. The dimensions of the systems with the solid and ventilated disc are also presented in Figure 4a,b, respectively. In order to verify the obtained theoretical temperature values, corresponding experimental data were used, obtained on a full-scale dynamometer at the Railway Research Institute in Warsaw, Poland (Table 2). The temperature of the ventilated disc was measured using thermocouples in accordance with the UIC (International Union of Railways) Leaflet 541-3. The rotational velocity of the disc was adjusted for the wheel of a rail vehicle with a diameter of $R_{w}=890\;\mathrm{mm}$. The six thermocouples $T_{n},\;n=1,2,\ldots ,6$ were situated symmetrically 1 mm below the friction surface on both sides of the ventilated disc (Figure 4c). The radial distances from the axis of rotation of the disc to the individual measurement points were equal to r=207 mm for ${\hat{T}}_{1}$ and ${\hat{T}}_{2}$, r=247 mm for ${\hat{T}}_{3}$ and ${\hat{T}}_{4}$, and r=287 mm for ${\hat{T}}_{5}$ and ${\hat{T}}_{6}$. In the circumferential direction, the distance between successive measurement points was 120°.

Numerical simulations and experimental measurements with the use of thermocouples were performed for two (no. 1 and no. 2) brake applications with different, experimentally recorded operating parameters (Table 2). This table contains the values of the input parameters, such as simulated energy per braking system, brake cylinder filling time $t_{1}$ and braking times $t_{s}$, with their respective braking distances $l_{1}$ and $l_{s}$, forces $F_{1}$ and $F_{s}$, and angular velocities $\omega_{0}$, $\omega_{1}$ and $\omega_{s}$. Table 2 also includes the temperature values at the point corresponding to the location of the thermocouple $T_{6}$, determined experimentally $T_{6,exp}$ and obtained numerically $T_{6,\mathrm{theor}}^{I}$, $T_{6,\mathrm{theor}}^{\mathrm{II}}$ and $T_{6,\mathrm{theor}}^{\mathrm{III}}$ on the basis of the three above-mentioned FE calculation models. The point corresponding to the position of the thermocouple $T_{6}$ was chosen because of the highest temperature reached at this point compared to the temperature at the other five points. In the case of a solid disc, the relative percentage difference between the values $\left|T_{6,\mathrm{theor}}^{\mathrm{III}}-T_{6,\mathrm{theor}}^{\mathrm{II}}\right|$ is 0.2% and 1% for braking applications no. 1 and no. 2, respectively. The relative difference between the values of the temperature of the ventilated disc, determined theoretically $T_{6,\mathrm{theor}}^{\mathrm{III}}$ and experimentally $T_{6,exp}$, is fully acceptable and is equal to 14.4% (no. 1) and 3.8% (no. 2).

The time profiles of the angular velocity ω(t) (Equation (2)) and the angular distance ϑ(t) (Equation (20)), obtained with the input parameters from Table 2 for the two variants of braking are presented in Figure 5. The corresponding temperature changes in time at a selected point P(−197 mm,208 mm) on the disc friction surface are shown in Figure 6. The temperature evolutions obtained on the basis of models III (solid disc) and II (ventilated disc) are marked in red and blue, respectively. These evolutions consist of a series of cycles (oscillations), each of which comprises two stages: first increase, and then decrease of the temperature. The first stage relates to the situation when the measuring point is still inside the nominal contact area during the movement of the pad, and the second—when the point P is already outside the contact area of the pad with the disc.

Due to the lower initial velocity $\omega_{0}$ and shorter braking time $t_{s}$ (Table 2), the number of cycles in the case of braking no. 1 (Figure 6a) is smaller than during braking no. 2 (Figure 6b). With each new cycle during the first stage $0\leq t<t_{1}$ of the braking process, corresponding to a linear increase in pressure, the maximum temperature increases, suffers a slight decrease at $t_{1}=4.2\;s$ as the pressure force increases abruptly from value $F_{1}$ to nominal $F_{s}$, and then increases again until the time corresponding to approximately half of the braking time ($t≅0.5t_{s}$). After this, a decrease in the maximum temperature is observed with each subsequent cycle, lasting until the end of the process $t=t_{s}$. It should be noted that such a time course of temperature during braking is determined by the time profile of the friction power density q (Equation (10)), including the stages increasing with the start of braking, reaching the maximum value, and the next reduction to the end time $t=t_{s}$. During braking no. 1, the maximum temperature of the ventilated disc (113.22 °C) is practically the same as the temperature of the solid disc (113.07 °C). For braking no. 2, the maximum temperature values were 170.47 °C and 169.61 °C, respectively. At the end of the braking process $t=t_{s}$, the temperature of the ventilated disc was 76.96 °C and for the solid disc 74.83 °C (braking no. 1), whereas for braking no. 2 it was 110.14 °C and 104.83 °C, respectively.

The oscillation amplitudes of the temperature time profiles decrease along with the distance from the disc friction surface, and at the depth above 5 mm they disappear altogether (Figure 7). During both brake applications, the temperature values determined with the use of the computational models of the solid and ventilated disc are very similar. Therefore, it can be concluded that the temperature distribution in the circumferential direction in both the solid and the ventilated disc at this depth is homogeneous. Therefore, the heating of the ventilated disc can be modeled using the simplification consisting of the fact that a layer with a thickness of about 5 mm rotates, and the remaining area of the disc towards its interior may be stationary. It would be far easier to simulate the rotation of the disc in relation to stationary pads, using for this purpose only the basic tools of the FEM programs, adapted in the COMSOL package. The available Translational Motion tool requires only the velocity field, with no additional disc deformation options. The obtained maximum temperatures at the point $P_{6}(-197\;\mathrm{mm},208\;\mathrm{mm},-1\;\mathrm{mm})$ corresponding to the location of the thermocouple $T_{6}$ for the ventilated and solid discs are, respectively, 96.7 °C and 96.5 °C during braking no. 1 and 147.2 °C and 145.8 °C during braking no. 2. These values are collated in Table 2 with the experimental data measured by the thermocouples.

The temperature distribution along the thickness of the disc for the selected time moments at point P are presented in Figure 8. For both brake applications, at the initial stage of the process, lasting approximately 6 s, the differences in temperature values of the solid and ventilated discs are insignificant and are observed at the depths greater than 10 mm. After 10 s, these differences appear on the friction surface and increase when moving away from it. The biggest temperature difference between the solid and ventilated discs occurred at the depth |z|=18 mm at the time moment t=14 s and was equal to 5.5 °C (Figure 8a) and 10.4 °C (Figure 8b) for braking no. 1 and no. 2, respectively.

The evolutions of the work W (Equation (13)) done during braking are shown in Figure 9. Increasing monotonically with time, at the moment of stopping, the calculated values of the friction work are equal to 989.4 kJ (Figure 9a) and 2096 kJ (Figure 9b) for braking no. 1 and no. 2, respectively. These values agree well with the relevant experimental data contained in Table 2, concerning the initial kinetic energy of the system $W_{0}$ (985.8 kJ, 2085 kJ). Such compliance additionally confirms credibility of the obtained numerical results.

Based on Equations (16) and (17), the changes in the braking time of the heat $W_{p,d}^{diss}$ dissipated to the environment from the free surfaces of the pad and the disc according to two mechanisms—convection and radiation—are shown in Figure 10 and Figure 11. For both brake applications, the heat $W_{p,d}^{rad}$ (Figure 10a and Figure 11a) dispersed to the environment by radiation is small compared to the heat $W_{p,d}^{conv}$ (Figure 10b and Figure 11b) transferred by convection, and even smaller compared to the work of friction W (Figure 9), converted into heat. Such small, almost negligible, amounts of energy dissipated due to thermal radiation were expected and are physically justifiable. The energy absorption from the surfaces of adjacent elements is of utmost importance. It should also be noted that the heat emitted as a result of radiation is reflected between the inner walls of the ventilated disc and is not dissipated to adjacent elements other than the disc. Therefore, significant amounts of thermal radiation energy are dissipated only through the exposed surfaces of the disc and partly internal and external cylindrical surfaces [11]. Despite the higher emissivity of the material of the pad than the disc material, the low thermal conductivity of the pad and its small free surface prevent it from playing a significant role in heat dissipation. In turn, the heat dissipated through convection from the free surfaces of the disc to the environment is small in relation to the total friction work during braking. The above shows that convection during a short-term single braking (from a few to several seconds) does not have any significant importance in the overall thermal balance of the brake. This explains, in particular, that the temperature of the ventilated disc is higher than that of the solid disc (Figure 6). As convection and radiation during short-term braking (from a few to several seconds) do not play a significant role in the overall thermal balance of the brake, thermal conductivity plays the main role in heat dissipation. The thickness of the material layer of the disc adjacent to the pad in a ventilated type of the disc is smaller than in a solid disc of the same mass. Hence, there is less heat dissipation by conduction in the ventilated disc.

The above conclusions are confirmed by the temperature distributions on the external surfaces of the ventilated disc and the solid disc of the same mass in selected time moments t=0.4 s, 2 s, 10 s, and 14.6 s during braking no. 1, as shown in Figure 12, Figure 13, Figure 14 and Figure 15. A slight difference of 0.1 °C between the maximum temperature value reached in the ventilated disc and the solid disc appears 2 s after the start of braking (Figure 13), and the greatest difference is 2.5 °C, i.e., about 3% of the maximum temperature value reached—at the end t=14.6 s (Figure 15). In terms of quality, the temperature distributions in both types of discs (solid and ventilated) are the same. The absence of any displacements of the heating area in the ventilated and solid discs results from the adapted angular displacement in time, determined independently before the temperature calculations of the COMSOL Multiphysics^®^ software. If this simulation step was omitted and all calculations were performed only in the COMSOL Multiphysics^®^ environment with a standard time step, slight differences in the position of the heating area would be observed and thus would also be seen in the temperature distributions.

## 6. Summary and Conclusions

Two calculation models of the ventilated disc temperature were developed, differing in their taking into account its rotational motion. For comparative purposes, in one of these models, the shape of the ventilated disc was simplified by replacing the pillars with an annular area, assuming that the new disc thus formed has the same mass as the ventilated disc. Based on the results of the simulation of the temperature mode during two single braking events (no. 1 and no. 2), differing in the initial kinetic energy, it was established that

The maximum temperature values obtained with the use of both numerical models are consistent with the corresponding results of measurements with thermocouples on the full scale inertial test bench;The differences in the maximum temperature values of the ventilated disc (model II) and the solid disc (models I and III) are negligible and amount to about 0.15 °C (no. 1) and 0.86 °C (no. 2). At the end of the process, these differences are 2.13 °C (no. 1) and 5.31 °C (no. 2);The amplitude of temperature oscillations caused by the relative motion of the pad and the disc with a linear increase in pressure is the highest in the middle of the braking process, and the oscillations themselves decrease with the distance from the friction surfaces and appear in the disc to a depth of about 5 mm;Temperature distribution in both discs indicates the possibility of disregarding the rotation of the disc material at a depth exceeding 5 mm. In our opinion, the result is significant, since it allows for a simple modeling of the rotation of the axisymmetric ring area of the disc located at a distance from the friction surface and allows the area below this depth to remain stationary;Even after several seconds of braking, the disc is not heated evenly across its entire thickness. At the end of the process, only the layer reaching one-fourth of the thickness of the analyzed ventilated disc is heated;The heat dissipation due to convection and thermal radiation is not significant in the total heat balance during braking;Unforced convection heat exchange with the environment during short-term braking does not lead to significantly better cooling of the ventilated disc as compared to a solid disc. This effect may become significant only after a few or several dozen seconds after stopping [25,26];The performed calculations justify the use of the constant heat transfer coefficient with a single short-term braking.

The obtained results, apart from the differences between the ventilated and the solid disc, may suggest possible discrepancies in the temperature fields in the case when the ventilated disc, combined with the irregularly shaped pad, is replaced with an axially symmetric disc brake model. Such a brake model should actually be two models, with separate heating of the disc and the pad. In the case of the disc, the cross-section of the solid disc created in this work should be replicated; there is no other possibility of replacing the ribs (pillars) of the ventilated disc. On the other hand, it would be difficult to reproduce the cross-section of the pad, which would be a model of the axisymmetric heating of the disc.

## Figures and Tables

**Figure 1 materials-14-07804-f001:**
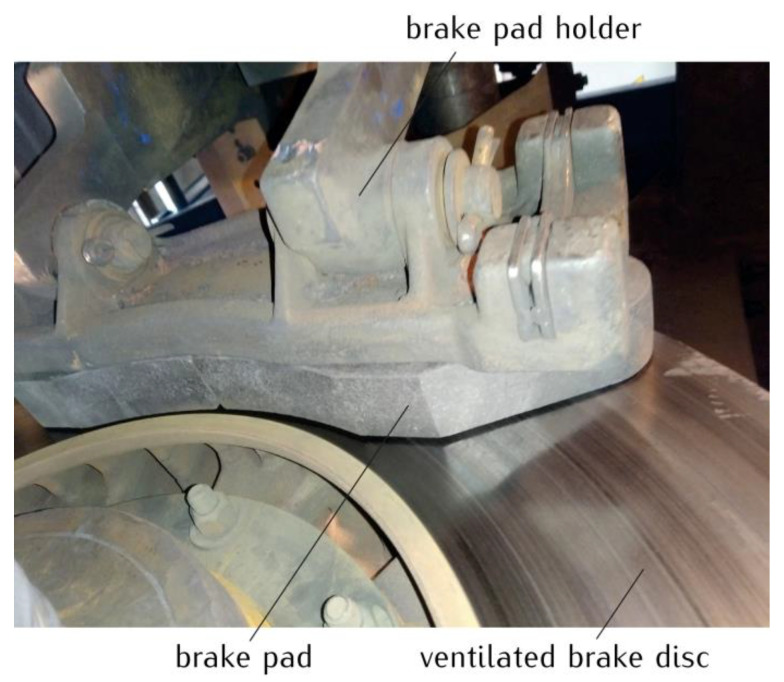
An example of a ventilated disc brake system of a railway vehicle mounted on the inertial test stand in the Railway Research Institute in Warsaw, Poland (picture courtesy of the Railway Research Institute in Warsaw).

**Figure 2 materials-14-07804-f002:**
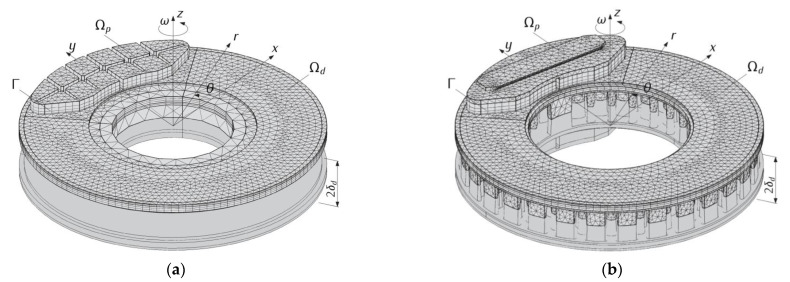
FE meshes and geometric 3D CAD models of the solid (model III) (**a**) and ventilated (model II) (**b**) types of disc brakes.

**Figure 3 materials-14-07804-f003:**
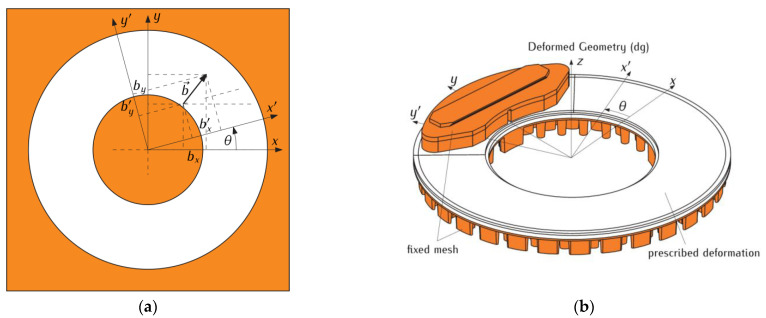
Motion scheme for the pad-disc system adapted in COMSOL Multiphysics^®^ software: (**a**) deformation of the layer of the disc in plane; (**b**) selection of the 3D objects of the braking system in Deformed Geometry (dg).

**Figure 4 materials-14-07804-f004:**
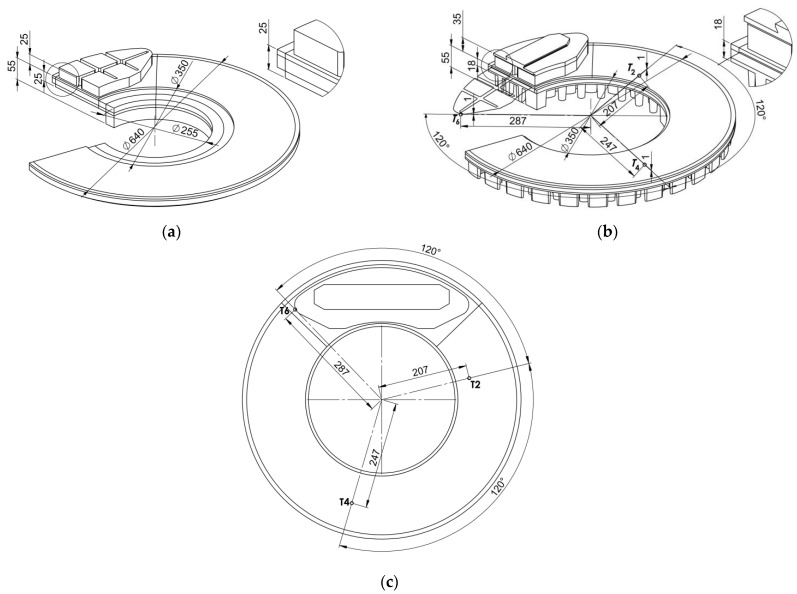
Dimensions of the solid (**a**) and the ventilated (**b**) disc brakes, with locations of the thermocouples (**c**).

**Figure 5 materials-14-07804-f005:**
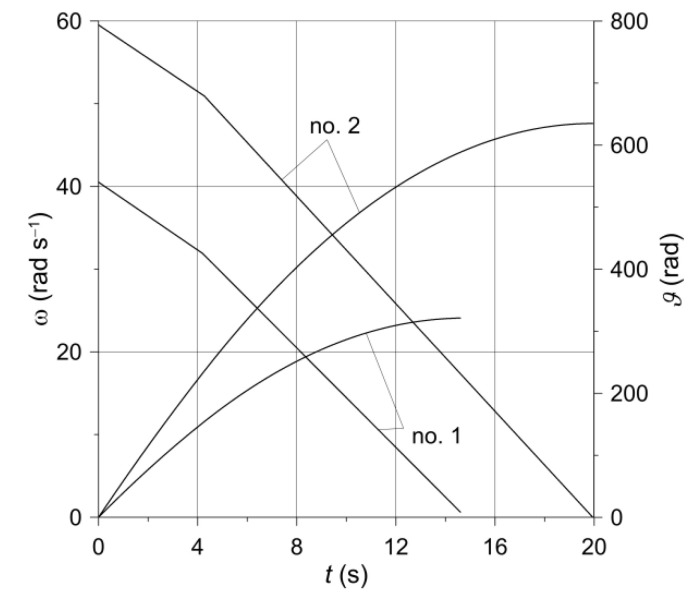
Changes in angular velocity ω(t) and angular distance ϑ(t) of the disc.

**Figure 6 materials-14-07804-f006:**
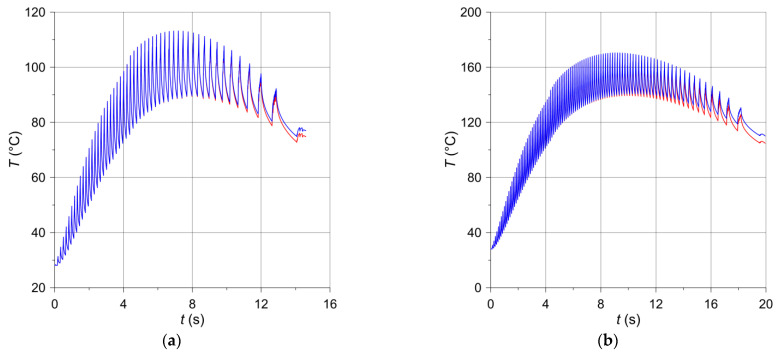
Temperature changes in time at location (x=−197 mm,y=208 mm,z=0 mm) for the solid (red lines) and the ventilated (blue lines) brake discs: (**a**) braking no. 1; (**b**) braking no. 2.

**Figure 7 materials-14-07804-f007:**
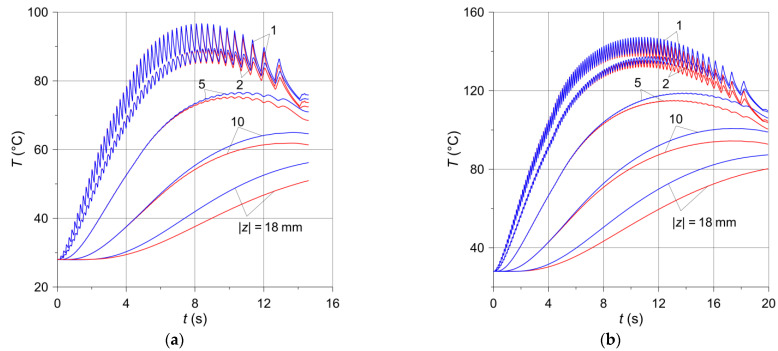
Temperature changes in time at location (x=−197 mm,y=208 mm) under the friction surface of the solid (red lines) and ventilated (blue lines) brake discs: (**a**) braking no. 1; (**b**) braking no. 2.

**Figure 8 materials-14-07804-f008:**
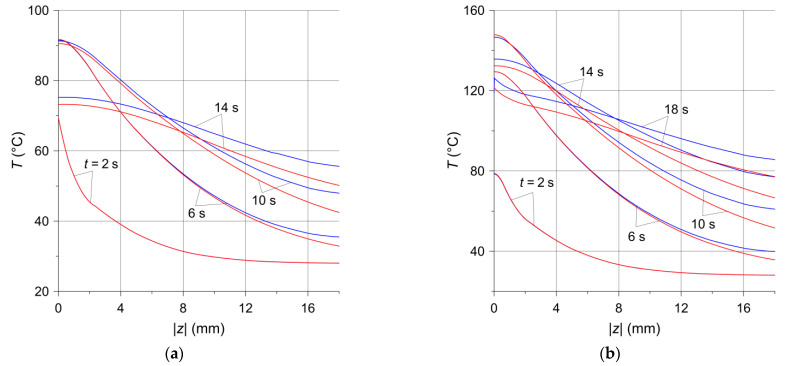
Temperature distributions in axial (z) direction of the solid (red lines) and ventilated (blue lines) disc at location (x=−197 mm,y=208 mm) from (**a**) braking no. 1; (**b**) braking no. 2.

**Figure 9 materials-14-07804-f009:**
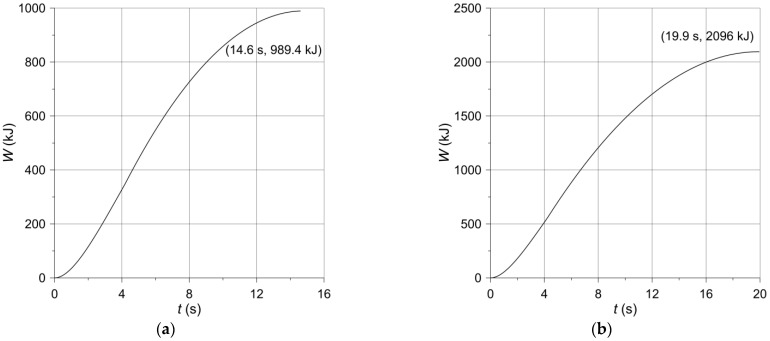
Kinetic energy W per one braking system converted into heat during (**a**) braking no. 1; (**b**) braking no. 2.

**Figure 10 materials-14-07804-f010:**
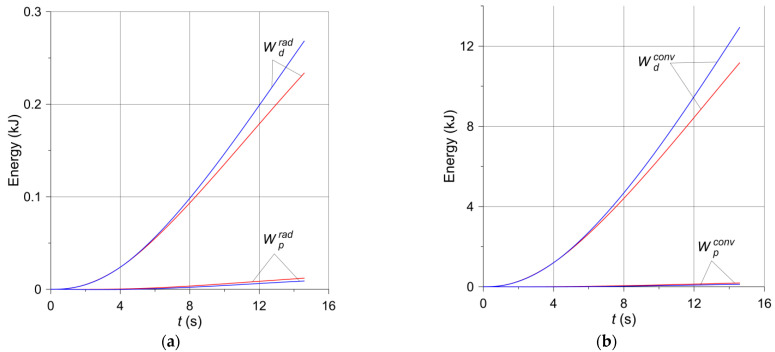
Changes in energy dissipated during braking no. 1 (solid disc—red lines, ventilated disc—blue lines) through (**a**) thermal radiation; (**b**) convection.

**Figure 11 materials-14-07804-f011:**
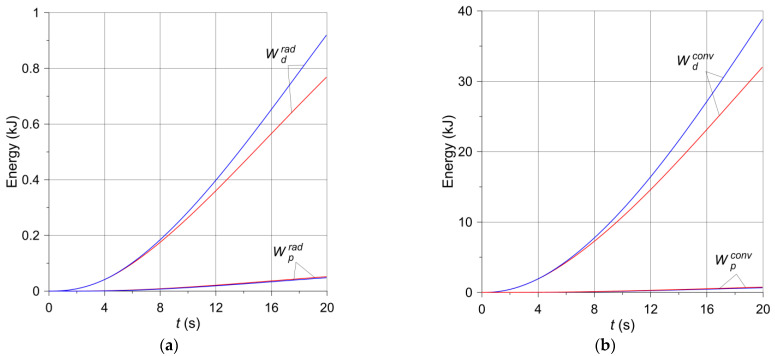
Changes in energy dissipated during braking no. 2 (solid disc—red lines, ventilated disc—blue lines) through (**a**) thermal radiation; (**b**) convection.

**Figure 12 materials-14-07804-f012:**
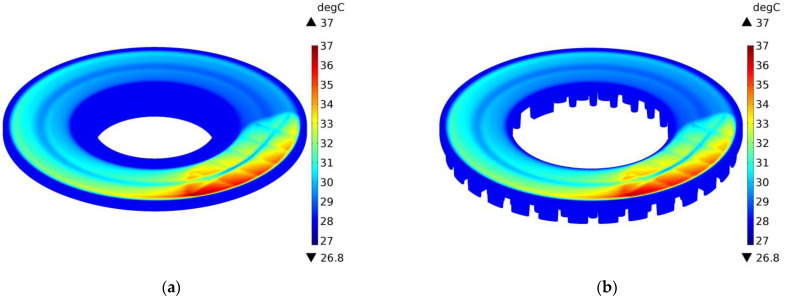
Temperature distribution in the solid (**a**) and ventilated (**b**) brake disc after t=0.4 s of braking no. 1.

**Figure 13 materials-14-07804-f013:**
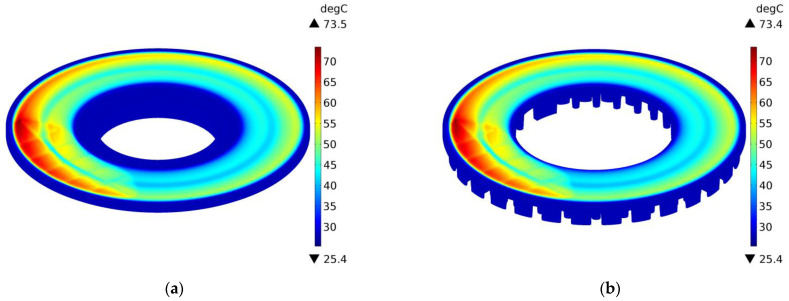
Temperature distribution in the solid (**a**) and ventilated (**b**) brake disc after t=2 s of braking no. 1.

**Figure 14 materials-14-07804-f014:**
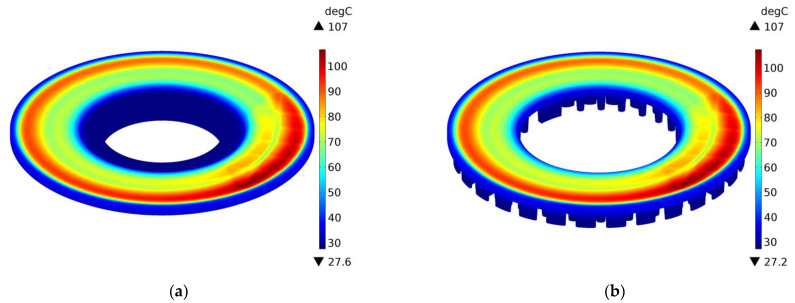
Temperature distribution in the solid (**a**) and ventilated (**b**) brake disc after t=10 s of braking no. 1.

**Figure 15 materials-14-07804-f015:**
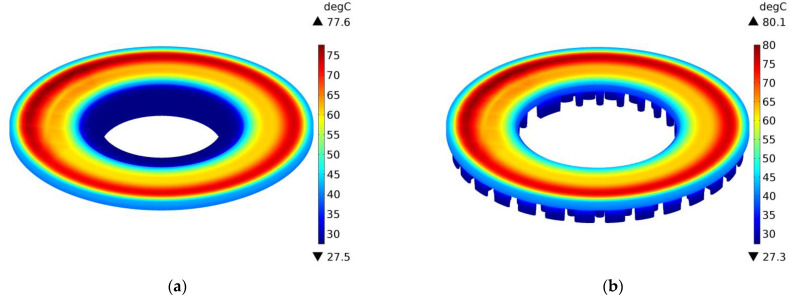
Temperature distribution in the solid (**a**) and ventilated (**b**) brake disc after t=14.6 s of braking no. 1.

**Table 1 materials-14-07804-t001:** Input parameters for numerical simulation.

Parameter	Disc	Pad
thermal conductivity, K($W\;m^{-1}\;K^{-1}$)	51	1.59
specific heat at constant pressure, c($J\;{\mathrm{kg}}^{-1}\;K^{-1}$)	500	770
density, ρ($\mathrm{kg}\;m^{-3}$)	7100	2450
surface emissivity, ε	0.28	0.8
heat transfer coefficient, h	100	100
initial temperature, $T_{0}$	28	28
thickness solid/ventilated, δ (mm)	55	25/35
outer radius, $R_{p,d}$ (mm)	320	303
inner radius, $r_{p,d}$ (mm)	175	178
inner radius of the hub solid/ventilated, $r_{p,d}$ (mm)	127.5/168.25	178
equivalent (friction) radius, $r_{eq}$ (mm)	-	247
surface area of contact of the pad on one side of the disc, $A_{a}$($m^{2}$)	-	0.034
surface area of the pad ($m^{2}$)	-	0.241
surface area the solid/ventilated disc ($m^{2}$)	1.287/2.002	-

**Table 2 materials-14-07804-t002:** Experimental data.

No.	$W_{0}$ (kJ)	$M_{b}$ (kg)	$V_{0}$ $(\mathrm{km}\;h^{-1}$)	$\omega_{0}$ $(\mathrm{rad}\;s^{-1}$)	$\omega_{1}$ $(\mathrm{rad}\;s^{-1}$)	$\omega_{s}$ $(\mathrm{rad}\;s^{-1}$)	$F_{1}$ (kN)	$F_{s}$ (kN)	f (m)	$l_{1}$ (m)	$l_{s}$ (m)	$t_{1}$ (s)	$t_{s}$ (s)	$T_{6,\exp}$ (°C)	$T_{6,\mathrm{theor}}^{I}$ (°C)	$T_{6,\mathrm{theor}}^{\mathrm{II}}$ (°C)	$T_{6,\mathrm{theor}}^{\mathrm{III}}$ (°C)
1	985.8	6066.7	64.9	40.5	31.9	0.6	47.1	49.3	0.31	71.7	143.4	4.2	14.6	84.5	96.5	96.7	96.5
2	2085.3	5951.5	95.3	59.5	50.9	0	47.5	49.5	0.31	179.8	286.5	4.2	19.9	153	145.8	147.2	145.8

The average values of the coefficient of friction f were calculated according to the card UIC 541-3.

## Data Availability

Not applicable.

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
