# Peer review of "Comparative Analysis of Temperature Fields in Railway Solid and Ventilated Brake Discs"

_materials, 2021, doi:10.3390/ma14247804_

Round 1

Reviewer 1 Report

  1. The sections 1 and 6 (“Introduction”, “Summary and Conclusions”) are too long and wordy, the authors had better condense them.
  2. Compared to the solid brake disc, the ventilated brake disc obtained the superior thermal properties (shown in Figures 10-11), but it showed the higher maximum temperature (shown in Figure 15), and why.
  3. From Table 2, we can find that the friction coefficients are 0.31 at braking energy of 985.8 kJ and 2085.3 kJ. Is it the average friction coefficient? The friction coefficient is exactly the same at different braking energy?
  4. Please provide the information of “mass per brake disc”, “friction radius” and “initial braking speeds with km/h”.
  5. Unforced convection heat exchange with the environment during short-term braking does not lead to significantly better cooling of the ventilated disc as compared to a solid disc”. Why not carry out the tests at higher braking speed or braking energy?

Author Response

We hereby would like to thank Reviewer for the favorable review and for providing constructive comments. The listed responses to comments are enclosed below. In the revised version of the manuscript, responses to reviewers’ comments are marked in red.

Comment 1:

The sections 1 and 6 (“Introduction”, “Summary and Conclusions”) are too long and wordy, the authors had better condense them.

Answer 1:

Our intention was to present as precisely as possible the current state of research on this subject and to highlight the problem discussed in the manuscript. The thermal problem of friction is so extensive that the presented scope is a necessary part directly related to the topic. However, section “Summary and Conclusions” as suggested, has been improved.

Comment 2:

Compared to the solid brake disc, the ventilated brake disc obtained the superior thermal properties (shown in Figures 10-11), but it showed the higher maximum temperature (shown in Figure 15), and why.

Answer 2:

More energy through convection and thermal radiation was dissipated from the ventilated disc due to the larger surface area (Fig. 10-11). In the case of the ventilated disc, the surface area was equal to 2,002m^2, while the surface area of the solid disc was only 1.287m^2. Obtaining a higher value of the maximum temperature on the friction surface for the ventilated disc resulted from the smaller thickness of the area just below the contact surface. The heat was dissipated more quickly from the friction surfaces by conduction, but was still accumulated in the volume of the brake disc. The bulk temperature of the ventilated disc was found to be lower than that of the solid disc. In order to better interpret the results, we added the values of the surfaces areas of the ventilated and solid discs to Table 1.

Comment 3:

From Table 2, we can find that the friction coefficients are 0.31 at braking energy of 985.8 kJ and 2085.3 kJ. Is it the average friction coefficient? The friction coefficient is exactly the same at different braking energy?

Answer 3:

Yes, these are the average values of the coefficient of friction calculated according to the formula from the UIC 541-3 card. We have included a relevant comment under Table 1.

Comment 4:

Please provide the information of “mass per brake disc”, “friction radius” and “initial braking speeds with km/h”.

Answer 4:

The quantity “mass per brake disc” and “initial braking speeds with km/h” were added to Table 2. The friction radius was already included in Table 1. It was called the equivalent radius.

Comment 5:

Unforced convection heat exchange with the environment during short-term braking does not lead to significantly better cooling of the ventilated disc as compared to a solid disc”. Why not carry out the tests at higher braking speed or braking energy?.

Answer 5:

Yes, it is a very interesting research topic that we are currently dealing with. Temperature measurements have already been made at higher initial speeds as well as with a higher mass per brake system. The obtained experimental data are analyzed.

Reviewer 2 Report

  • The introduction of this article is very well written, the topic and its analysis is thoroughly explained.
  • Half of the brake wheel/disc with one brake pad is analyzed – I consider this a good approach to the analysis. This is further explored and explained in the paper with the possibility of only rotating the top layer of the disc, rather than rotating the whole body, which further simplifies the analysis in software programme such as ANSYS.
  • This article could be well extended into closer analysis of e.g. drilled brake discs, grooved discs used in automotive industries etc..
  • Article very well describes the process of optimisation of the analysis.
  • Please be advised in the118-119 lines, you have mentioned 3 types of brake disc used and you have written just 2 types, this seems to be a mistake.
  • Figure 1 is quite small and has low readability. Increasing its size and making space between the figure and description (line) would improve readability and overall look.
  • The lines 342,457 and 484 need more space between them and figures
  • In some paragraphs minor spell check required
  • It is stated in the paper that ventilated discs have worse aerodynamic properties. Is this not a negligible amount?
  • Theoretical values of temperature are higher than experimental in the first application in table 2., while lower than experimental in aplication No.2, why is this so?
  • The results in fig 7 show that ventilated discs do reach higher or indifferent temperatures – this is an interesting and unexpected result, since the expectation is reaching lower temperatures in ventilated discs, rather than in solid ones. Could this expectation be reached by prolonging the time of braking – number of cycles?
  • It has been stated in the conclusion of the article that thermal conductivity plays the main role in heat dissipation. The article addresses the questions and informs the reader about the possibilities of simplification of brake disc analysis methods. This provides a valuable addition to the current methods of brake disc analysis.
  • The article was overall interesting to read, results are consistent. I believe the article is very much relevant today, and will be for the future too since the given analysis methods are still being used today in development of brake discs, and brake discs are undoubtedly going to be used in the long term future too.

Author Response

We hereby would like to thank Reviewer for the favorable review and for providing constructive comments. The listed responses to comments are enclosed below. In the revised version of the manuscript, responses to reviewers’ comments are marked in red.

Comment 1:

The introduction of this article is very well written, the topic and its analysis is thoroughly explained.

Answer 1:

Thank you for your favorable comment.

Comment 2:

Half of the brake wheel/disc with one brake pad is analyzed – I consider this a good approach to the analysis. This is further explored and explained in the paper with the possibility of only rotating the top layer of the disc, rather than rotating the whole body, which further simplifies the analysis in software programme such as ANSYS.

Answer 2:

Thank you for your positive and valuable comment.

Comment 3:

This article could be well extended into closer analysis of e.g. drilled brake discs, grooved discs used in automotive industries etc.

Answer 3:

This is a very good direction for further research. However, the main purpose of this article was to analyze the brake discs of railway vehicles for which we conducted experimental tests.

Comment 4:

Article very well describes the process of optimisation of the analysis.

Answer 4:

Thank you for your positive and valuable comment.

Comment 5:

Please be advised in the118-119 lines, you have mentioned 3 types of brake disc used and you have written just 2 types, this seems to be a mistake.

Answer 5:

In fact, we have misspelled the sentence. Thank you for paying attention. The mistake has been corrected.

Comment 6:

Figure 1 is quite small and has low readability. Increasing its size and making space between the figure and description (line) would improve readability and overall look.

Answer 6:

We put into manuscript a new photograph made at the Railway Research Institute in Warsaw, Poland.

Comment 7:

The lines 342,457 and 484 need more space between them and figures

Answer 7:

The comment has been taken into account.

Comment 8:

In some paragraphs minor spell check required.

Answer 8:

The comment has been taken into account.

Comment 9:

 It is stated in the paper that ventilated discs have worse aerodynamic properties. Is this not a negligible amount?

Answer 9:

Actually, we did not analyze the aerodynamic properties. As for the temperature level, these differences are insignificant at such a short distance analyzed in the study. We expect that they will be more noticeable in our current research on long-term and repeated braking. This information was an important step for us to decide on the choice of a calculation model.

Comment 10:

Theoretical values of temperature are higher than experimental in the first application in table 2., while lower than experimental in aplication No.2, why is this so?

Answer 10:

In our opinion, the differences of 10% for this type of tests are small and sufficient to reliably estimate the temperature of the disc during braking. The experimental temperature measurement is affected by imperfection of the measurement, including the lag of the thermocouples, the quality of contact of the thermocouple tip with the disc material, etc.

Comment 11:

The results in fig 7 show that ventilated discs do reach higher or indifferent temperatures – this is an interesting and unexpected result, since the expectation is reaching lower temperatures in ventilated discs, rather than in solid ones. Could this expectation be reached by prolonging the time of braking – number of cycles?

Answer 11:

The answer to this remark was given to Reviewer 1. Its content is presented below.

 “More energy through convection and thermal radiation was dissipated from the ventilated disc due to the larger surface area (Fig. 10-11). In the case of the ventilated disc, the surface area was equal to 2,002m^2, while the surface area of the solid disc was only 1.287m^2. Obtaining a higher value of the maximum temperature on the friction surface for the ventilated disc resulted from the smaller thickness of the area just below the contact surface. The heat was dissipated more quickly from the friction surfaces by conduction, but was still accumulated in the volume of the brake disc. The bulk temperature of the ventilated disc was found to be lower than that of the solid disc. In order to better interpret the results, we added the values of the surfaces areas of the ventilated and solid discs to Table 1.”

Other effects can be expected with longer braking times and with repeated braking applications.

Comment 12:

It has been stated in the conclusion of the article that thermal conductivity plays the main role in heat dissipation. The article addresses the questions and informs the reader about the possibilities of simplification of brake disc analysis methods. This provides a valuable addition to the current methods of brake disc analysis.

Answer 12:

Thank you for your favorable comment.

Comment 13:

The article was overall interesting to read, results are consistent. I believe the article is very much relevant today, and will be for the future too since the given analysis methods are still being used today in development of brake discs, and brake discs are undoubtedly going to be used in the long term future too.

Answer 13:

Thank you very much for the positive response to our work.